# An Exploratory Review on the Potential of Artificial Intelligence for Early Detection of Acute Kidney Injury in Preterm Neonates

**DOI:** 10.3390/diagnostics13182865

**Published:** 2023-09-05

**Authors:** Yogavijayan Kandasamy, Stephanie Baker

**Affiliations:** 1School of Medicine and Public Health, The University of Newcastle, Callaghan, NSW 2308, Australia; 2Department of Neonatology, Townsville University Hospital, Townsville, QLD 4814, Australia; 3College of Medicine and Dentistry, James Cook University, Townsville, QLD 4810, Australia; 4College of Science and Engineering, James Cook University, Cairns, QLD 4878, Australia; stephanie.baker@jcu.edu.au

**Keywords:** neural network, predictive algorithm, acute kidney injury, premature neonates

## Abstract

A preterm birth is a live birth that occurs before 37 completed weeks of pregnancy. Approximately 15 million babies are born preterm annually worldwide, indicating a global preterm birth rate of about 11%. Up to 50% of premature neonates in the gestational age (GA) group of <29 weeks’ gestation will develop acute kidney injury (AKI) in the neonatal period; this is associated with high mortality and morbidity. There are currently no proven treatments for established AKI, and no effective predictive tool exists. We propose that the development of advanced artificial intelligence algorithms with neural networks can assist clinicians in accurately predicting AKI. Clinicians can use pathology investigations in combination with the non-invasive monitoring of renal tissue oxygenation (rSO_2_) and renal fractional tissue oxygenation extraction (rFTOE) using near-infrared spectroscopy (NIRS) and the renal resistive index (RRI) to develop an effective prediction algorithm. This algorithm would potentially create a therapeutic window during which the treating clinicians can identify modifiable risk factors and implement the necessary steps to prevent the onset and reduce the duration of AKI.

## 1. Introduction

Preterm birth is a live birth before 37 completed weeks of pregnancy. Approximately 15 million babies are born preterm annually worldwide, indicating a global preterm birth rate of about 11%. With 1 million children dying due to preterm birth before the age of five, it is the leading cause of death among children under five years [1]. Two-thirds of human nephrons, the functional units in the kidney, develop during the third trimester, and nephrogenesis is complete by 36 weeks of gestation [2]. No new nephrons develop in infancy or adult life, meaning premature birth before 29 weeks GA significantly impairs nephron endowment [2].

Clinicians use GFR (glomerular filtration rate) to measure kidney function, and this value can be estimated (eGFR) using serum creatinine (SCr) [3]. SCr is a product of skeletal muscle metabolism, excreted almost entirely via the kidneys [3]. However, SCr measurement, especially in neonates, is associated with many pitfalls: (a) there is a significant delay in the rise of SCr after the renal insult (48–72 h); (b) SCr only starts to rise once GFR falls by 50% [3]; (c) it is influenced by maternal SCr level [4]; (d) in premature neonates, the plasma SCr level increases in the first 48 h and then declines [5]; (e) changes due to sex and size may influence levels [6]; (f) the presence of analytical interferents in neonatal samples, such as jaundice and haemolysis, cause preanalytical errors using the Jaffe reaction [6]; and (g) the SCr level depends on muscle mass [6].

AKI is defined as an increase in the SCr level and a reduction in the 24 h urine output (Neonatal Modified Kidney Disease Improving Global Outcomes (KDIGO) consensus) [7]. AKI is associated with an increased mortality risk-adjusted odds ratio of 4.6 [95% confidence interval 2.5–8.3; *p* < 0.0001] [8]. Data from a large, multicentre international (AWAKEN) study showed that nearly 50% of neonates ≤ 29 weeks GA admitted to a neonatal intensive care unit (NICU) developed AKI [8]. Renal hypoperfusion (caused by bleeding, patent ductus arteriosus (PDA) and infection) leading to renal hypoxia, together with the reduced nephron endowment secondary to prematurity, are critical in the pathogenesis of AKI [9,10]. Commonly used essential neonatal medications such as aminoglycosides, amphotericin, acyclovir, diuretics, and nonsteroidal anti-inflammatory drugs (NSAIDs) (for the treatment of PDA) are nephrotoxins and further damage the immature kidneys [11]. Beyond the risk of AKI, nephrotoxin exposure directly and permanently affects the developing kidney and nephrogenesis [12]. Several animal studies have shown that the nephron number is reduced if neonates are exposed to gentamicin or indomethacin [13,14,15]. During life, conditions such as hypertension, glomerulonephritis and diabetes further impact renal function and increase the likelihood of developing CKD [16]. In a systematic review, we identified 31 human studies investigating the short-term and long-term kidney outcomes of prematurity. We concluded that prematurity is likely linked to an increased risk of kidney dysfunction and hypertension in childhood and early adulthood [17]. We have also shown that prematurely born neonates have smaller kidneys (and therefore a reduced nephron number), but normal eGFR compared with term neonates [18,19]. Since the eGFRs were similar, premature neonates must have a higher single-nephron GFR. This suggests that they are already hyperfiltering, thus laying the foundations for early adult life nephron loss through glomerulosclerosis [18,19]. SCr and urine outputs are the only biomarkers available for clinicians to diagnose AKI using the KDIGO classification. There are no proven treatments for established AKI and no effective predictive tools [10]. The development of an effective predictive tool is critical for creating a therapeutic window, and thus it is the first step towards improving outcomes for patients with AKI.

This exploratory review contributes to the literature by identifying key technologies that are strong candidates for developing a predictive tool. The role of artificial intelligence in helping renal clinicians to improve patient outcomes is gaining attraction. Nephropathologists can use artificial intelligence to better diagnose renal pathology, predict prognosis, and provide therapy responsiveness from kidney biopsies [20,21,22]. As computing power increases exponentially, it is envisaged that artificial intelligence’s role in diagnosis and management will only increase. In this review, firstly, we examine the recent literature on non-invasive techniques for assessing renal function in the neonatal cohort. We then explore the use of artificial intelligence for processing data from non-invasive sensors to identify AKI early. Our findings indicate that artificial intelligence (AI) is a strong candidate for predicting AKI onset using non-invasive sensors and, thus, is a promising avenue for future research. 

## 2. Methods and Materials

Our literature search was conducted using PubMed, Scopus, and Web of Science for relevant studies. Our keywords included “non-invasive”, “acute kidney injury”, “AKI”, “artificial intelligence”, “neural network”, and “machine learning”. Initial searches led to more specific searches on technologies of interest, including “near-infrared spectroscopy”. Studies that focused on the non-invasive measurement of renal function or the use of AI to diagnose or identify early biomarkers of AKI in any age cohort were considered. Studies that focused on renal function metrics that require pathological or invasive testing were excluded, as were studies that focused on using AI to predict the prognosis of patients with existing AKI rather than early diagnosis. Non-English papers were also excluded. In terms of AI papers, we focused on papers published between 2017 and 2023 due to the fast-moving nature of this field and its related technologies. 

The first outcome of this review was to identify several non-invasive tools for assessing renal function and the renal function metrics that can be obtained from these tools. The second key outcome was to investigate the suitability of AI for developing a predictive algorithm for AKI onset, using metrics obtained from non-invasive devices. 

## 3. Results

### 3.1. Non-Invasive Assessment of Renal Function

#### 3.1.1. Renal Artery Doppler Ultrasound 

Renal artery Doppler is clinicians’ most common non-invasive method to quantitatively assess renal vascularity and perfusion [23,24]. Renal perfusion is responsible for glomerular filtration, and although the renal artery resistance index (RRI) is not currently part of KDIGO guidelines, it is often used as a measure of renal perfusion and a nonspecific prognostic marker for various disorders that affect the kidney [25]. Doppler waveforms not only reflect blood flow velocities, but the derived RRI gives information regarding the pattern evolution of flow over time and is calculated using the following formula [24]:RRI = (peak systolic velocity (PSV) − end diastolic velocity (EDV))
(peak systolic velocity (PSV))

The early detection of a change in the RRI can predict alterations in GFR before any change in SCr is observed [26]. The normal range in infants is 0.50–0.70 [21,22]. High resistive indices (>0.8) are associated with renal dysfunction. In one study, the sensitivity of RRI in predicting AKI was 71–74%, with a specificity of 46–74% (area under the curve: 0.60–0.75) [27]. 

#### 3.1.2. Near-Infrared Spectroscopy (NIRS)

Adequate renal perfusion and oxygenation are of critical importance in neonates. NIRS is an emerging non-invasive cot-side monitoring tool that can identify inadequate tissue oxygenation in premature neonates [28]. NIRS provides clinicians with an estimate of local tissue oxygen utilisation by assessing post-capillary oxygenation (rSO_2_). NIRS offers a feasible, non-invasive approach to the continuous monitoring of renal oxygenation over time, serving as a surrogate marker for renal perfusion [29]. Multiple factors may affect NIRS values, but the two main determinants are tissue perfusion and tissue oxygen utilisation [30]. NIRS provides rSO_2_ values; these values depend on GA at birth and chronological/corrected age in neonates. All premature neonates in the NICU have ongoing continuous oxygen saturation monitoring (SpO_2_) as part of their routine clinical care. Using rSO_2_ and SpO_2_ values, we can calculate the oxygen consumption of renal tissue (rFTOE) using the formula [30,31]:rFTOE = (SpO_2_ − rSO_2_)/SpO_2_

Studies have shown an association between reduced rSO_2_ and rFTOE levels and impaired kidney function, and increased SCr in term neonates and those undergoing surgery [29,32]. However, this technology is currently not routinely used for monitoring kidney function in premature babies in Australia or internationally, and remains a research tool [11].

### 3.2. Artificial Intelligence for AKI Prediction

Artificial intelligence (AI) models can be used to establish links between data and outcomes. They have therefore seen significant use in the healthcare domain recently, particularly in measuring health signs or predicting outcomes from non-invasive sensors [33]. However, the application of AI to NIRS data has not been broadly explored; to the best of our knowledge, it has not yet been considered for AKI predictions from NIRS data.

One recent study [34] developed an AI model that utilised vital signs, laboratory results, and other clinical information to train an AI model for predicting AKI in a paediatric cohort. The model was able to predict AKI onset 30 h in advance, achieving an area under the receiver-operator curve (AUROC) of 0.89. This indicates a good ability to distinguish between AKI and non-AKI patients. However, the model is limited by its dependency on parameters obtained via laboratory testing, which are time-consuming for clinical staff and can cause discomfort for patients—particularly during repeated measurement. Thus, a model that can predict AKI from non-invasive sensor data is preferable in the clinical context.

One recent study considered this, and demonstrated that rSO_2_ and rFTOE measurements could be linked with the onset of AKI in a neonatal cohort using a logistic regression approach [32]. Their findings indicated that rSO_2_ > 70% was linked to an AKI onset at 48 h of life with an area under the receiver-operator curve (AUROC) of 0.73 and sensitivity of 84%. Additionally, RFTOE ≤ 25 was shown to be predictive for AKI at 54–66 h of life, with AUROC values ranging between 0.8 and 0.83. This predictive performance is reasonable; however, it could be significantly improved by considering multiple features simultaneously.

Advanced AI models such as neural networks and ensemble models are likely to have much stronger predictive performance due to their ability to consider many input parameters and their relationships simultaneously; however, such models have not yet been explored for predicting AKI from non-invasive NIRS data.

Candidate models for AKI prediction from clinical features and time-series rSO_2_ and rFTOE data include long short-term memory (LSTM) models and other recurrent neural network (RNN) structures, as these have been broadly used for interpreting time-series data [31]. Previously, LSTM has been used to interpret data extracted from NIRS sensors to assess bladder fullness in patients living with neurogenic bladder dysfunction, achieving promising results [35]. Another potential candidate is convolutional neural networks (CNNs), which are a strong candidate for interpreting raw data from Doppler ultrasound and NIRS due to their strong ability to identify key features in image and waveform data [31]. Hybrid models that combine RNN and CNN structures may also be suitable, as these have previously shown a strong ability to identify important features in raw data as well as the relationships between these features [36]. Another candidate outside neural network structures is random forest (RF), an ensemble model architecture comprising many decision trees. RF was previously used to predict chronic kidney disease from discrete measurements, including vital signs and metrics obtained from urine and blood tests [37]. Novel AI architectures are continuously emerging; thus, exploring newer architectures, such as transformers, is also warranted.

AI approaches have significant potential for developing predictive algorithms for AKI onset using data from non-invasive sensors. However, this application of AI has not been explored in the literature to date. Thus, a significant research opportunity remains in developing AI tools for early AKI prediction using Doppler ultrasound and NIRS data. This would provide a non-invasive solution that creates a therapeutic window in which clinicians may be able to intervene to improve patient outcomes.

In addition to developing AI models that can predict AKI, explainable artificial intelligence techniques can also be utilised to understand biomarkers associated with AKI. Explainable AI techniques, including Shapley additive explanations (SHAP) [38] and local interpretable model-agnostic explanations (LIME) [39], can be utilised to identify which features most strongly contributed towards the accurate prediction of AKI. Previous studies have used explainable artificial intelligence tools to assess which health parameters are most strongly linked with a particular outcome [40,41]. In one work, SHAP was used to understand vital sign biomarkers associated with mortality in neonatal intensive care, identifying that a high variability in heart rate and respiratory rate are strongly linked with mortality [40]. Additionally, the AI identified low gestational age as a key factor in neonatal mortality.

Given that this is a well-established relationship in the literature, it increases confidence that the AI model focuses on suitable features [40]. In another work, LIME and SHAP were used to identify risk factors associated with AKI in an adult cohort who underwent cardiac surgery, providing novel insights into biomarkers for AKI onset [41]. Thus, it is likely that explainable artificial intelligence tools will strongly support the identification of NIRS parameters that are predictive of AKI onset (Figure 1), which may guide the development of treatments for AKI.

### 3.3. Outcomes for Clinical Practice

Based on our literature review, we envisage cot-side non-invasive kidney monitoring becoming part of routine cot-side monitoring for premature neonates in the NICU, similar to how cardiac and respiratory monitoring is currently provided for all premature neonates. Early research suggests that NIRS metrics measured up to 48 h in advance can be linked with AKI onset using simple logistic regression [29]; however, there remains a significant opportunity for developing advanced algorithms that consider multiple metrics to predict AKI onset.

Based on prior research in the field of AI for healthcare [31], we anticipate that AI algorithms could be trained to accurately identify “at-risk” premature neonates who are likely to develop AKI days in advance, creating a therapeutic window for clinician intervention (Figure 2). Furthermore, such a system would be able to continue learning and improving itself with time and more data sets. This would see AI models become more accurate over time and ensure that improvements in treatment and care are reflected in their knowledge. We further anticipate that explainable AI tools could be readily applied to the developed AI systems, helping to identify biomarkers associated with AKI whilst simultaneously enhancing clinician trust in AI decisions.

As a result of predictive AI models, clinicians would be able to recognise modifiable risk factors and implement the necessary steps to alleviate AKI risk, such as choosing antibiotics that are metabolised in the liver instead of the kidneys, fewer toxic medications for kidney function (for example, using paracetamol in the treatment of PDA instead of ibuprofen), and providing premature neonates with more suitable total parenteral nutrition (TPN) support to maintain growth adequately. There is also a significant opportunity to leverage the findings made by AI to support research into novel treatments for AKI in the future.

## 4. Conclusions

In this exploratory review, we have examined the literature on the use of non-invasive sensors for assessing renal kidney function and investigated how the data obtained from these sensors can be leveraged to predict AKI onset in a neonatal cohort. Our findings indicated that NIRS is a promising tool for measuring renal function non-invasively, with links recently being made between NIRS parameters and AKI onset in neonatal cohorts. We additionally identified that AI architectures, including LSTM, CNN, and RF, are strong candidates for the improved prediction of AKI, using NIRS data based on related work in the literature to date. Explainable AI was also identified as a key technology for identifying biomarkers and ensuring that AI systems are accountable to and interpretable by the clinicians who depend on them. Overall, this review reveals the need for further research into the use of responsible AI to predict AKI onset using non-invasive data in the neonatal cohort. Such a tool would create a valuable therapeutic window, contribute to understanding AKI, and significantly improve patient outcomes.

## Figures and Tables

**Figure 1 diagnostics-13-02865-f001:**
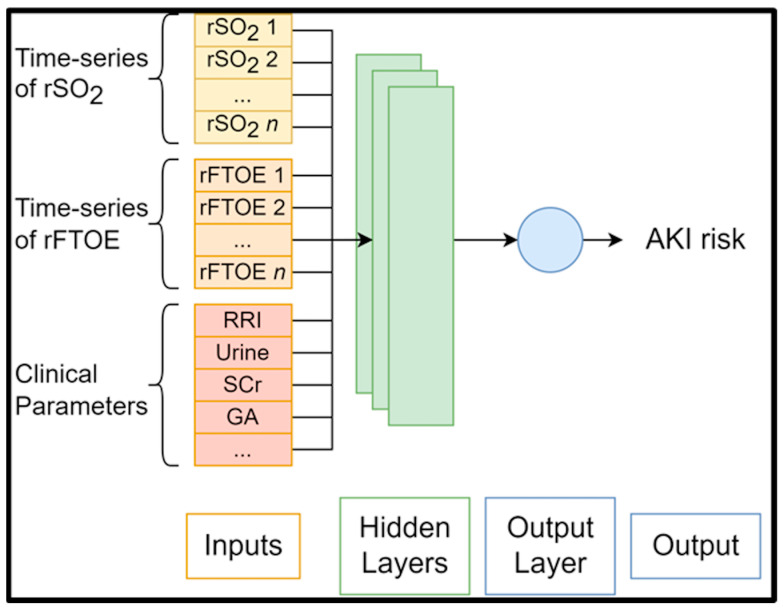
Generic structure of AI algorithms for AKI risk prediction.

**Figure 2 diagnostics-13-02865-f002:**
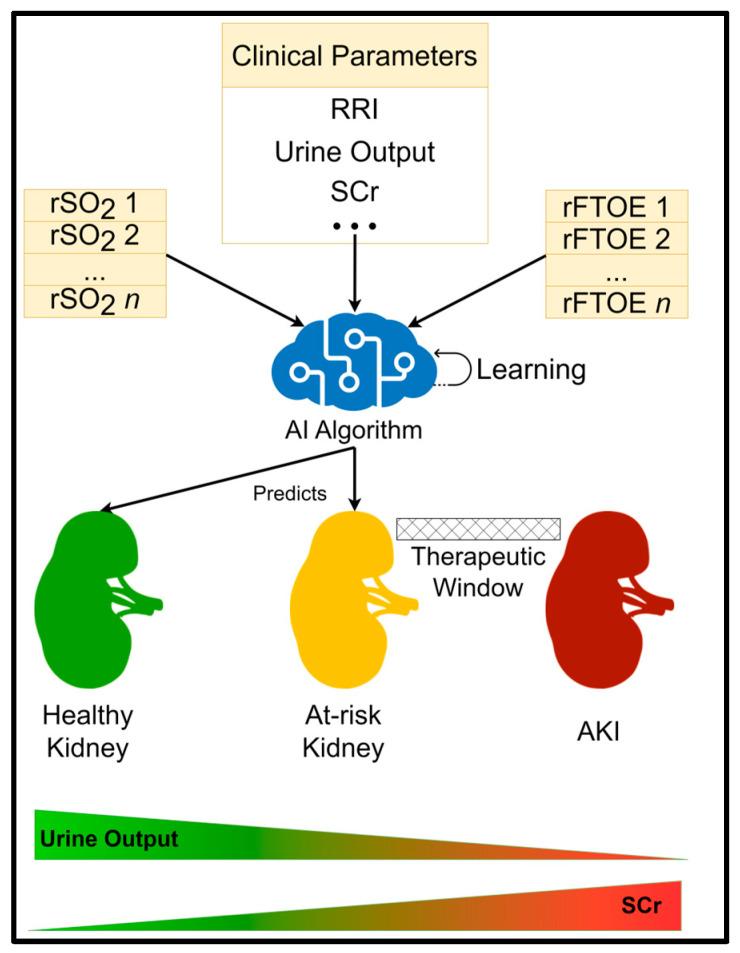
AI prediction system overview.

## Data Availability

Data sharing not applicable.

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
