# Peer review of "An Exploratory Review on the Potential of Artificial Intelligence for Early Detection of Acute Kidney Injury in Preterm Neonates"

_diagnostics, 2023, doi:10.3390/diagnostics13182865_

Round 1

Reviewer 1 Report

Authors propose that the development of advanced artificial intelligence algorithms with neural networks can assist clinicians in accurately predicting AKI. Clinicians can use pathology investigations in combination with noninvasive monitoring of renal tissue oxygenation (rSO2) and renal fractional tissue oxygenation extraction (rFTOE) using near-infrared spectroscopy (NIRS) and renal resistive index (RRI) to develop an effective prediction algorithm. This algorithm would potentially create a therapeutic window during which the treating clinicians can identify modifiable risk factors and implement the necessary steps to prevent the onset and reduce the duration of AKI. 

The paper is well written and the design and results are clearly. Can you briefly discuss the role of invasive (vs non invasive) tools in this diagnostic. It is undoubtoful that non invasive tools should be the diagnostic gold standard but the role of expertise and artificial intelligence in the interpretatation of kidney biopsies are a milestone and  dederve a discussion or introduction.

Please quote:

Histopathology. 2021 May;78(6):791-804. doi: 10.1111/his.14304. Epub 2021 Mar 8.

J Nephrol. 2022 Sep;35(7):1801-1808. doi: 10.1007/s40620-022-01327-8. Epub 2022 Apr 19.

Kidney Int. 2020 Jul;98(1):65-75. doi: 10.1016/j.kint.2020.02.027. Epub 2020 Apr 1.

Author Response

We thank the reviewer for the comments.

Can you briefly discuss the role of invasive (vs non invasive) tools in this diagnostic. It is undoubtoful that non invasive tools should be the diagnostic gold standard but the role of expertise and artificial intelligence in the interpretatation of kidney biopsies are a milestone and  dederve a discussion or introduction.

Reply: We have provided a brief discussion on the use of AI in kidney biopsies in the introduction and have included all 3 suggested references.

Thank you

Reviewer 2 Report

This study addresses the worldwide frequency of premature births and its relationship with the development of acute kidney injury (AKI) in premature infants. He proposes that advanced artificial intelligence algorithms (in combination with neural networks) can help clinical specialists to accurately predict AKI. However, no conclusions about the proposed model were included in the study. In the results section, no appropriate result of any of the models compared could be included for comparison. For this reason, there are many shortcomings of the study that cannot be corrected in a short time.

1- The resolution of Figure 1 should be increased and added to the study again.

2- What does the "Nil" mentioned in the Acknowledgements and Funding statements mean?

3- No numerical results of the proposed model were included in the study.

4- The proposed model is said to be better than models such as LSTM, CNN and RF, but no results based on them were included in the study.

5- There is no information about the data used in the study.

6- Information such as numerical data from previous studies and the methods used could not be found in the literature section.

Author Response

This study addresses the worldwide frequency of premature births and its relationship with the development of acute kidney injury (AKI) in premature infants. He proposes that advanced artificial intelligence algorithms (in combination with neural networks) can help clinical specialists to accurately predict AKI. However, no conclusions about the proposed model were included in the study. In the results section, no appropriate result of any of the models compared could be included for comparison. For this reason, there are many shortcomings of the study that cannot be corrected in a short time.

Reply: The purpose of this Brief report is to demonstrate the paucity of data in this area, and the need for further research. The last 2 sentences in our manuscript reflect this view.

  • The resolution of Figure 1 should be increased and added to the study again.

Reply: We have increased the resolution as advised for Figure 1

  • What does the "Nil" mentioned in the Acknowledgements and Funding statements mean?

Reply: We have revised this section

  • No numerical results of the proposed model were included in the study.

Reply: We do not have any data to report at this stage. We are in the process of carrying out a research project and hope to report it in due course.

  • The proposed model is said to be better than models such as LSTM, CNN and RF, but no results based on them were included in the study.

Reply: This is merely our proposal/hypothesis. Once our actual study data is available, we will evaluate different models.

  • There is no information about the data used in the study.

Reply: As the reply for 3

  • Information such as numerical data from previous studies and the methods used could not be found in the literature section.

Reply: We have included 3 additional references as suggested by reviewer 1. There is no data in the literature on using AI to predict AKI in premature neonates. We are in the process of carrying out a research project and hope to report it in due course, in a different manuscript.

Reviewer 3 Report

In this review Kandasamy and Baker go over how the field of nephrology can use AI to help diagnose AKI in neonates at bedside and in real time. The authors do a good job identifying the areas that make the current diagnostic tools such as Serum Creatinine measurements not sensitive and specific enough and explain how other factor during the newborn period, child and adult life can impact kidney function and further compromising it.

They continue to explain the importance of AI and how it can help the field in predicting AKI. The authors explain how a literature search was done to look for non-invasive techniques that can be used to predict AKI using AI and if AI has been already used in the field.

The authors do a strong case on why AI can help, and how different AI models have been previously used in the field and how they can be improved. There is less explanation on why this will be specifically helpful to the neonatal population, whether this is due to the fact that some of the noninvasive techniques are not used in the adult population and why those techniques commonly used in the adult population are not good in neonates and why.

Both figures do a good job summarizing how the authors propose how AI works on processing data to predict AKI.

A brief paragraph explaining general result before going into the specifics may help introduce the reader into them.

Overall, this is an important contribution that tries to summarize current AI models used to predict AKI and what factor will be important to consider when trying to improve their performance in predicting AKI in neonates.

Author Response

Thank you for the comments. We don't have anything further to add.

Round 2

Reviewer 2 Report

I have been excitedly reviewing the necessary corrections in your work. But unfortunately, the lack of concrete input on this issue and the presence of many studies in the field of medicine will overshadow your work. According to your suggestion /hypothesis, unfortunately, I still can't make sense of your defense of superiority over models such as LSTM, CNN and RF. This suggestion/hypothesis should be supported by scientific data. Even if this is a research article, I do not consider it appropriate for you to make such a suggestion/hypothesis. If you do not have enough concrete results at the moment, it would be more appropriate to write your study in the style of a research article examining the studies in this field and defend it.

Author Response

Thank you for the comment. This is a review article and reflects the authors' view. We don't have any data at the moment but plan to carry out a study in the future.